# Rules of engagement between αvβ6 integrin and foot-and-mouth disease virus

Abhay Kotecha[1], Quan Wang[2], Xianchi Dong[3,4], Serban L. Ilca[1], Marina Ondiviela[5], Rao Zihe[2,6], Julian Seago[7], Bryan Charleston[7], Elizabeth E. Fry[1], Nicola G.A. Abrescia[5,8,*], Timothy A. Springer[3,4,*], Juha T. Huiskonen[1,*] & David I. Stuart[1,9,*]

Foot-and-mouth disease virus (FMDV) mediates cell entry by attachment to an integrin receptor, generally αvβ6, via a conserved arginine–glycine–aspartic acid (RGD) motif in the exposed, antigenic, GH loop of capsid protein VP1. Infection can also occur in tissue culture adapted virus in the absence of integrin via acquired basic mutations interacting with heparin sulphate (HS); this virus is attenuated in natural infections. HS interaction has been visualized at a conserved site in two serotypes suggesting a propensity for sulfated-sugar binding. Here we determined the interaction between αvβ6 and two tissue culture adapted FMDV strains by cryo-electron microscopy. In the preferred mode of engagement, the fully open form of the integrin, hitherto unseen at high resolution, attaches to an extended GH loop via interactions with the RGD motif plus downstream hydrophobic residues. In addition, an N-linked sugar of the integrin attaches to the previously identified HS binding site, suggesting a functional role.

[1] Division of Structural Biology, The Nuffield Department of Medicine, University of Oxford, The Henry Wellcome Building for Genomic Medicine, Headington, Oxford OX3 7BN, UK. [2] National Laboratory of Macromolecules, Institute of Biophysics, Chinese Academy of Sciences, Beijing 100101, China. [3] Program in Cellular and Molecular Medicine and Division of Hematology, Department of Medicine, Boston Children's Hospital, Boston, Massachusetts 02115, USA. [4] Department of Biological Chemistry & Molecular Pharmacology, Harvard Medical School, Boston, Massachusetts 02115, USA. [5] Structural Biology Unit, CIC bioGUNE, CIBERehd, 48160 Derio, Spain. [6] Laboratory of Structural Biology, School of Medicine, Tsinghua University, Beijing 100084, China. [7] Pirbright Institute, Pirbright GU24 0NF, UK. [8] IKERBASQUE, Basque Foundation for Science, 48013 Bilbao, Spain. [9] Diamond Light Sources, Harwell Science and Innovation Campus, Didcot OX11 0DE, UK. * These authors contributed equally to this work. Correspondence and requests for materials should be addressed to T.A.S. (email: springer@crystal.harvard.edu) or to J.T.H. (email: juha@strubi.ox.ac.uk) or to D.I.S. (email: dave@strubi.ox.ac.uk).

FMDV, a small non-enveloped RNA virus (genus *Aphthovirus*, family *Picornaviridae*)[1], exists as seven antigenically distinct serotypes and many variants and sub-types[2]. Serotype O poses the most serious global threat and is present in >80% of vaccines. Structures of several serotypes determined at near atomic resolution[3–7] have revealed a capsid composed of 60 copies each of four structural proteins (VP1 to VP4). A prominent feature of this otherwise rather smooth capsid is an exposed flexible loop (~30 residues) on VP1, termed the GH loop. This loop and the RGD motif embedded within it (residues 145–147 in the O serotype) comprise a major site of neutralization by the host humoral immune response and mediate FMDV binding integrin[8–12].

Integrins are heterodimeric adhesive glycosylated membrane proteins found on the surface of most cells, capable of signalling in both directions through the plasma membrane. Both α and β subunits contain a large multi-domain extracellular N-terminal portion, a single trans-membrane helix and a small C-terminal cytoplasmic domain. Domains of both the α and β chain come together to form a ligand binding head, each connected through a leg to the membrane[13]. The four integrins identified as FMDV receptors *in vitro* (αvβ1, αvβ3, αvβ6 and αvβ8) have been previously identified to bind ligands at a small cleft at the subunit interface of the head. Specific recognition of RGD is achieved by binding of the α-subunit β-propeller domain to the Arg, while the Asp completes the MIDAS $Mg^{2+}$ cation site in the integrin β subunit βI domain[14,15]. We would expect the FMDV RGD motif to bind similarly. Integrin αvβ6 shows the highest affinity for FMDV in cell cultures, and binding induces virus delivery to early and recycling endosomes[11]. Unlike αvβ3, αvβ6 has additional specificity for hydrophobic residues downstream of the RGD and in most serotypes of FMDV, leucine residues L148 and L151 at D+1 and D+4 are key to interactions with αvβ6 and αvβ8 (refs 10,15).

Integrins are used by many viruses as a receptor for cell entry[16] but, as for FMDV, the flexibility of the integrin binding portions of the virus has made it difficult to visualize the interaction[17–19]. Indeed the only detailed structural information on RGD interactions with αvβ6 comes from a complex with a small RGD-containing pro-TGFβ peptide[15].

FMDV can often readily adapt to tissue culture, where infection can occur in the absence of integrin via acquired basic mutations (for example, H56R in VP3), which interact with sulfated sugars such as heparin sulfate (HS); this virus is attenuated in natural infections and we have previously visualized, in two serotypes, HS interaction at a conserved site suggesting that FMDV may have an underlying propensity for binding sulfated sugars[6,20,21].

Prompted by advances in cryo-electron microscopy (cryo-EM)[22] and computational methods for the analysis of flexible assemblies[23], we investigate here the interaction of recombinant αvβ6 with two intact FMDV particles representative of serotype O: O PanAsia (PanAsia; for which we also determined a high-resolution structure by X-ray crystallography) and O1 Manisa (O1M; whose structure was determined previously, PDB: 5AC9), both chemically inactivated (Methods). We visualize the integrin engaging the virus in a fully open conformation. The VP1 GH loop, to which it attaches via interactions with the RGD motif and downstream hydrophobic residues, is extended up away from the virus surface. A further attachment occurs via an N-linked sugar of the integrin attaching to the previously identified HS binding site.

## Results

### Localized reconstruction reveals distinct αvβ6 binding poses.

Data were collected for complexes of virus and mammalian expressed integrin constructs[15]. Initially for both viruses 2 mM $Mg^{2+}$ was included in the incubation mixture. Subsequently, for O1M only, we tested the effect of 2 mM $Mn^{2+}$. This resulted in more complete decoration of the O1M particles than with $Mg^{2+}$ and the $Mn^{2+}$ data were therefore used in further analysis. For all experiments, the micrographs clearly showed bound integrin and no evidence of capsid dissociation (Supplementary Fig. 1). Conventional single particle analysis assuming icosahedral symmetry (Methods), yielded density maps for the complexes with the virus capsids at 3.1 Å resolution for PanAsia (13,438 particles) and 3.5 Å for O1M (1,649 particles; Fig. 1a,b; Supplementary Figs 2 and 3, Table 1). In both cases the integrin density was visible but very blurred. Comparison of the PanAsia virus capsid with a high-resolution X-ray structure determined here of native PanAsia (Fig. 1c; Methods; Table 2) confirmed that the integrin had not significantly altered the virus structure, except for the GH loop of VP1, which contains the RGD (Fig. 1d). Although the icosahedrally averaged density for the GH loop and integrin displayed little detail (Fig. 1a,b), localized reconstruction enabled us to deconvolute distinct modes of integrin attachment, despite the variable occupancy of the integrin[23,24] (Methods). This method allows the independent analysis of all 60 potential binding sites (termed sub-particles) on each of the particles, and their respective classification into 3D classes in Relion (Methods). For both viruses only two of the 20 classes considered were well-defined, and these were taken forward for further analysis. Only a small fraction of integrin sub-particles was assigned to these particular classes. It is possible that some of the unassigned sub-particles correspond to different poses of the integrin that were not detected in our classification. The sub-particles also unavoidably overlap each other in the projection image, further hampering their 3D classification[23]. Likely due to these limitations, integrin density was resolved to a lower resolution than the capsid. For each virus the two well-resolved classes corresponded to distinct poses of the integrin, for PanAsia these were termed A and A′ and for O1M, A and B (Fig. 2a–d). The integrin portion for these complexes was resolved at 10.8 Å (pose A, 4.9% of sub-particles) and 11.5 Å (pose A′, 4.2% of sub-particles) for PanAsia and at 8.6 Å (pose A, 5.9% of sub-particles) and 12.3 Å (pose B, 6.7% of sub-particles) for O1M (resolution assessment uses the 'gold standard' protocol[24]). The remainder of the sub-particles were either undecorated (75% for PanAsia and 38% for O1M) or could simply not be assigned to one of these well-defined poses (15.9% for PanAsia and 49.4% for O1M). Decorated particles had between 5 and 55 integrins attached and statistical analysis showed no indication of the distribution being skewed by binding to neighbouring symmetry-related capsid sites (Methods; Supplementary Fig. 4). By composite modelling using prior structures (Methods) we established reliable models for all four poses (Fig. 3a–d). The integrin adopts the open, high-affinity conformation of the headpiece with the hybrid domain swung out by 75° (state 8) relative to its orientation in the closed state (Supplementary Fig. 5)[13,25,26], a universal mechanism of integrin activation also seen in EM with αvβ6 bound to pro-TGF-b1 (ref. 27).

In each of the structures, the orientation of the VP1 GH-loop is unambiguous (Fig. 4a,b,d). The GH-loop is pulled away from the virus surface, raising the RGD peptide by ~20 Å above its position in the reduced O1BFS virus where the GH loop packs stably on the virus surface[8]. This is achieved by rotations around residues at the base of the loop (*circa* residues 134 and 157; Fig. 1d). In most O serotype viruses, including PanAsia, a disulphide bond bridges residues 134 of VP1 and 130 of VP2, causing the loop to become disordered. Reduction of the disulphide bond stabilizes the loop against the virus surface abrogating

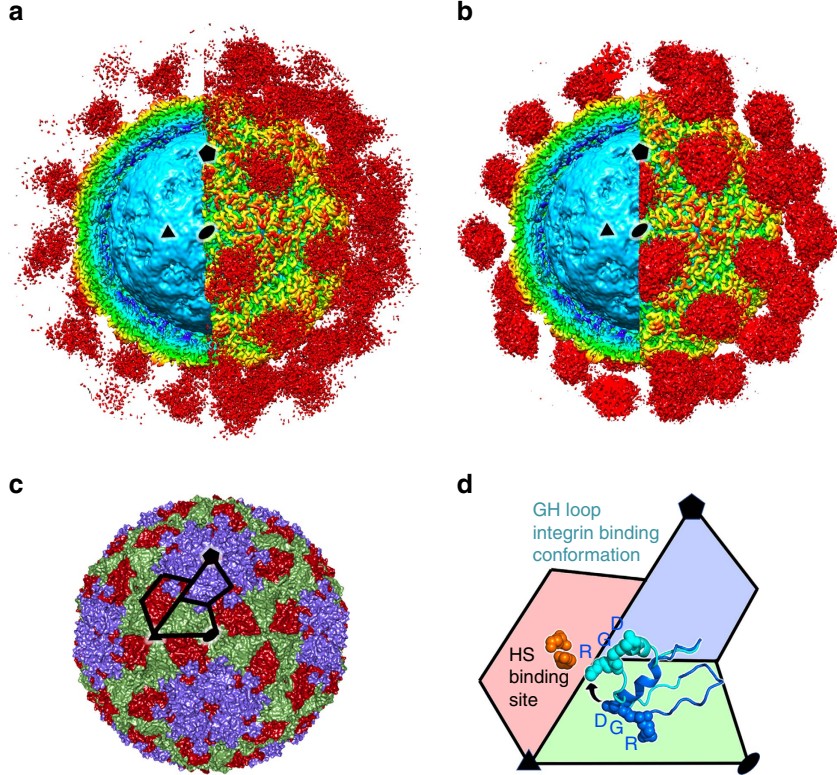

**Figure 1 | Apo and holo αvβ6-FMDV complexes.** (**a**) The density for the αvβ6-PanAsia complex determined by cryo-EM. On the left of the image the front half of the density has been cut away so that a cross-section of the binding can be seen. Depth-cueing is used such that colour indicates radius (<110 Å: cyan; 120–140 Å: yellow; >150 Å red). Clouds of red density show the bound integrin. (**b**) EM structure of αvβ6-O1M, depicted as for **a**. (**c**) The structure of PanAsia determined by crystallography. Colouring reflects the different viral proteins, VP1 (blue), VP2 (green) and VP3 (red). One asymmetric unit is outlined. (**d**) Enlargement of one asymmetric unit coloured as in **c**. Two conformations of the VP1 GH loop are rendered using a ribbon representation and the RGD motif is depicted with spheres. The conformational change from that seen in reduced O1 virus (PDB:1FOD) (blue) to the integrin bound conformation (cyan) is depicted with an arrow. Some of the residues involved in heparin sulfate (HS) binding are shown as orange spheres and the binding site is labelled. In all panels, one 2-fold, 3-fold and 5-fold axis of icosahedral symmetry has been labelled by a black ellipse, triangle and pentagon, respectively, and all panels show the same view down a 2-fold axis of symmetry.

## Table 1 | Cryo-EM data collection, reconstruction and refinement statistics.

| | O PanAsia | O1M |
|---|---|---|
| *Data collection and reconstruction* | | |
| Voltage (kV) | 300 | 300 |
| Movies | 360 | 285 |
| Frames | 25 | 25 |
| Dose rate (e⁻ per pixel per s) | 8 | 8 |
| Pixel size (Å) | 1.35 | 1.35 |
| Total dose (e⁻ Å⁻²) | 18 | 18 |
| Defocus (µm) | 1.5–3.0 | 1.5–3.0 |
| Particles* | 13,483 | 1,649 |
| Resolution (Å)† | 3.1 | 3.5 |
| B-factor (Å²)‡ | −120 | −92.9 |
| | | |
| *Model refinement* | | |
| Fo-Fc correlation | 0.84 | 0.82 |
| Protein atoms | 5,149 | 5,149 |
| R.m.s.d., bonds (Å) | 0.01 | 0.01 |
| R.m.s.d., angles (°) | 0.95 | 0.77 |
| Clash score, all atoms (percentile) | 9.45 | 11.53 |
| Rotamer outliers (%) | 0.0 | 0.0 |
| Ramachandran outliers (%) | 0.0 | 0.0 |

R.m.s.d., root mean squared deviation.
*The number of particles used in the final reconstruction is given.
†Resolution as estimated by Fourier shell correlation with 0.143 threshold.
‡B-factor used for map sharpening.

integrin binding[8]. Unusually, in O1M a serine is present at 134 of VP1, although the loop is nonetheless disordered on the native virus particle.

Integrin binding in pose A is very similar for both PanAsia and O1M (Fig. 2a,c); superposition of the integrin based on the virus structure shows <8° rotation between the two. In this conformation, the integrin molecule is almost perpendicular to the surface of the virus, as might be expected for initial engagement at the crowded cell surface (Fig. 5a). The slightly different binding mode on PanAsia (pose A′, Fig. 2b) shows the integrin 'moved up' and rotated 12° away from the virus surface whilst the second binding mode on O1M (conformation B, Fig. 2d) shows the integrin rotated by 158° around the vector normal to the virus surface at the point of attachment, to bring three integrin hybrid, PSI and EGF1 domains together around the icosahedral 3-fold axes. These modes of interaction demonstrate the extreme flexibility in the presentation of the RGD peptide.

At this resolution the cryo-EM density for the integrin does not reveal detailed interactions; however, there is density consistent with the RGD in the FMDV GH loop binding at the interface of the αv and β6 integrin subunits. The TGF-β peptide (sequence RGDLXXL) in the integrin-peptide complex forms a helix somewhat similar to that observed for the FMDV GH loop residues 148–156 (ref. 8, and, as bound to the β6 subunit[15], gives an excellent fit into the FMDV loop density (pose A; Fig. 4; Supplementary Fig. 5). This suggests a similar interaction of the RGDLXXL/I motif in FMDV via the RGD and leucines at +1

**Table 2 | X-ray Data collection and refinement statistics.**

| | O PanAsia* |
|---|---|
| *Data collection* | |
| Space group | I23 |
| Cell dimensions | |
| $a, b, c$ (Å) | 345.01, 345.01, 345.01 |
| $\alpha, \beta, \gamma$ (°) | 90, 90, 90 |
| Resolution (Å) | 50–2.3 (2.38–2.30)† |
| $R_{merge}$ | 0.27 (0.88) |
| $I/\sigma I$ | 3.1 (0.86) |
| Completeness (%) | 76.9 (45.9) |
| Redundancy | 2.1 (1.7) |
| | |
| *Refinement* | |
| Resolution (Å) | 50–2.3 |
| No. reflections | 229,892 |
| $R_{work}/R_{free}$ | 22.1/23.0 |
| No. atoms | |
| Protein | 5149 |
| Ligand/ion | — |
| Water | 110 |
| B factors | |
| Protein | 20.75 |
| Ligand/ion | — |
| Water | 24.54 |
| r.m.s. deviations | |
| Bond lengths (Å) | 0.006 |
| Bond angles (°) | 1.455 |

R.m.s., root mean square.
*Data collected from 60 different crystals were merged.
†Values in parentheses are for highest-resolution shell.

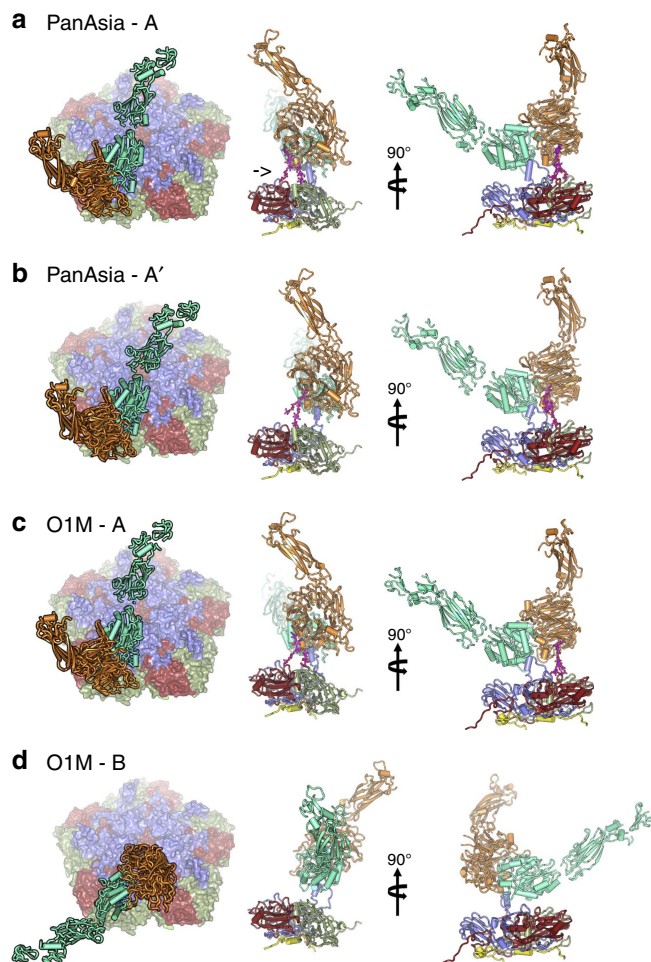

**a** PanAsia - A

**b** PanAsia - A′

**c** O1M - A

**d** O1M - B

**Figure 2 | αvβ6-FMDV binding modes.** Two predominant binding modes are shown for PanAsia (**a**,**b**) and for O1M (**c**,**d**). The left panel shows a view down onto the capsid surface (just a pentamer of each virus is shown colour-coded as in Fig. 1c). The integrin is drawn in cartoon representation with the alpha-helices rendered as rods; αv is orange and β6 is green. The right hand panels depict orthogonal side-views of the complexes. The integrin is drawn as in the left panel interacting with a protomer of the virus in cartoon style and colour coded as in Fig. 1c with VP4 in yellow. The FMDV VP1 GH loop, in blue, can be seen interacting with the integrin. The αv N-linked sugar, which forms an additional attachment to the virus, is drawn in magenta (marked with an arrow in **a**). The similarity in the binding mode 'A' between the two viruses is evident.

and +4 (Fig. 5b). The D of the RGD appears to coordinate the MIDAS $Mg^{2+}$ in the PanAsia complex and as expected the structure is similar in the O1M complex with $Mn^{2+}$ (Fig. 5c). Superimposition of the FMDV O1BFS reduced GH loop[8] onto the TGF-β peptide suggests that a switch in the backbone conformation between the D of the RGD and the subsequent L residue would allow the leucines to interact with the β6 subunit (Supplementary Fig. 5b), by rotating the helix some 90° about its long axis. At this resolution we cannot judge if any residues upstream of the RGD (139–140) interact with residues in the αv subunit conferring further specificity/stability.

**An unexpected glycan-virus interaction.** Unexpectedly, in pose A, additional density was observed bridging the integrin to the virus surface (for both PanAsia and O1M). This density locates ~29 Å away from the root of the GH loop (Fig. 4a; Supplementary Figs 6 and 7)[28] and corresponds to an N-linked sugar at N266 of the αv β-propeller domain) reaching to the HS binding site on the virus[6,20] (Figs 4c,e and 5b,d; Supplementary Figs 5–7). The quality of the density and lack of density in virus-only maps, rules out this being adventitious attachment of HS. The significance of this interaction remains unclear since both viruses examined are tissue culture adapted and therefore bear mutations in this region, and the glycosylation of the recombinant integrin $(Man5GlcNAc2)^{29}$ may differ from that seen in the course of natural infection. Nevertheless, this may recapitulate an underlying sugar–protein interaction site in field strains of the virus, which might explain the ease with which FMDV adapts to cell culture by increasing the basic charge in this region to create affinity for HS[21].

## Discussion

We have used high-resolution cryo-EM and a powerful localized reconstruction method to overcome the challenges in visualizing an inherently flexible attachment site. This reveals that FMDV extends its VP1 GH loop upwards and engages its integrin receptor in an open active state. The interaction mode of the GH loop with integrin replicates that described for the TGF-β peptide underlining a universal binding mechanism of the integrin to RGD-containing ligands except that the expected physiological integrin conformation is now observed.

Further, the integrin makes a glycan attachment to the virus in the vicinity of the HS binding site. This may explain the repeated generation of HS binding sites by simple charge-based amplification of a pre-existing sugar binding site. Detailed mapping of the interactions between αvβ6 and FMDV may ultimately facilitate the design of novel inhibitors of viral entry. More generally we expect such approaches to deconvoluting flexible modes of interaction using cryo-EM to have broad applicability within biology, not least to other virus receptor interactions, including other picornavirus–integrin complexes.

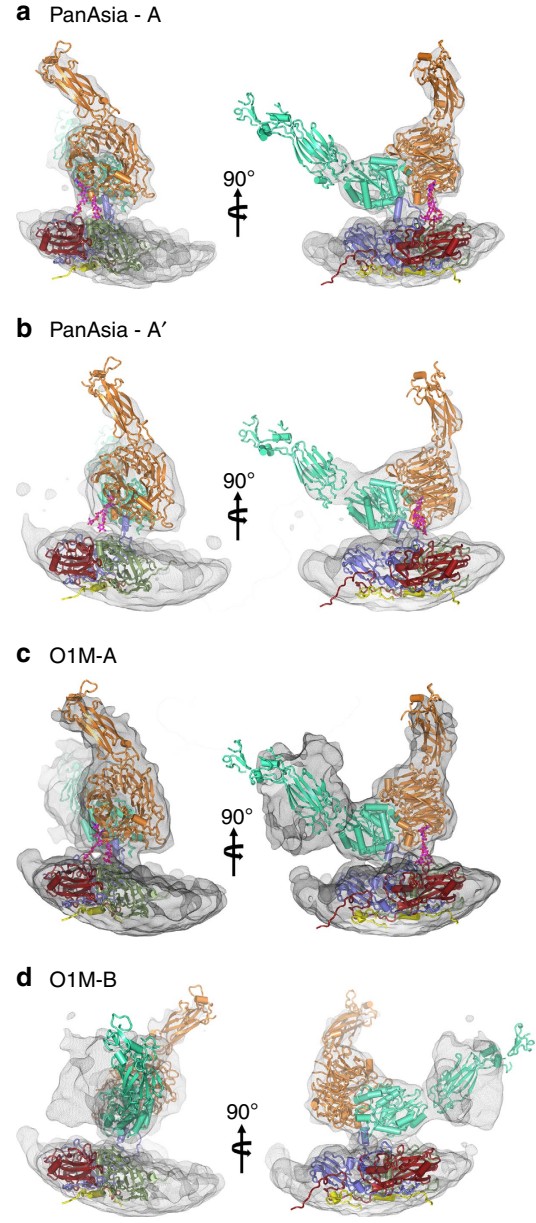

**Figure 3 | αvβ6–FMDV binding modes.** (**a–d**) The two predominant binding modes are shown for O PanAsia (**a,b**) and for O1M (**c,d**) as depicted in Fig. 2 but enlarged to show the electron density for the integrin component.

## Methods

**Virus production and purification.** Infectious FMDV O1K/O1Manisa (O1M) chimeric clones were constructed using reverse genetics by removing cDNA encoding the VP2, VP3, VP1 and 2A proteins from a derivative of the pT7S3 O1K infectious clone, termed pT7SBmuts, leaving cDNA encoding the Lpro, VP4, 2B, 2C, 3A, 3B, 3C and 3D proteins. The removed cDNA was replaced with the corresponding O1M cDNA from pGEM9zf subclones (UKG/35/2001; GenBank accession no. AJ539141)[30], RNA was transcribed from the infectious clones using the MEGAscript T7 kit (Invitrogen) and transfected into BHK-21 cells (Sigma (order # 85011433), BHK 21 (Clone 13) from hamster). After 24 h, the cells were frozen then thawed in their growth media. Following clarification by centrifugation, the supernatant containing the initial virus stock (termed 'passage 0', P0) was harvested. BHK-21 cells were subsequently used to passage the virus. Cells were infected for 24 h between passages. Lysate containing virus particles was inactivated by two consecutive incubations with binary ethyleneimine, following procedures in-line with disease security regulations at The Pirbright Institute. The lysate was stored at − 80 °C until required. For purification, inactivated virus particles were thawed at room temperature and precipitated by the addition of 8% w/v PEG 6000 by incubation at 4 °C overnight. Precipitated virus was harvested by centrifugation at 3,500g for 1 h at 4 °C, resuspended in 50 mM HEPES

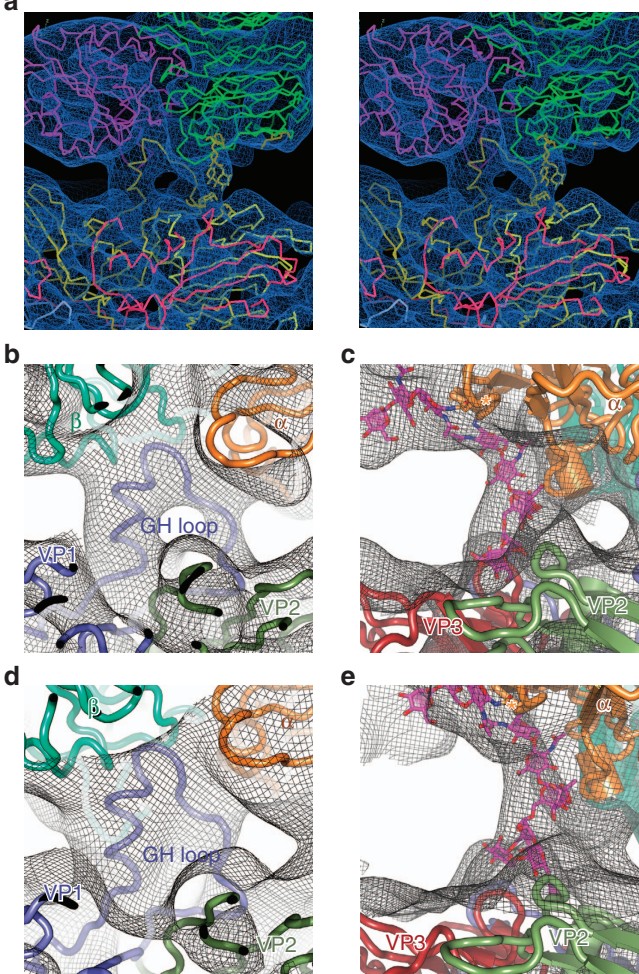

**Figure 4 | Close-up of mode A binding of αvβ6 to FMDV.** (**a**) overview in wall-eye stereo of the binding to O PanAsia (**b–e**) close-ups of the GH loops and N-linked carbohydrate density are shown for O PanAsia (**b,c**) and for O1M (**d,e**).

(4-(2-hydroxyethyl)-1-piperazineethanesulfonic acid), pH 8.0, containing 400 mM NaCl and 1% v/v NP-40, and clarified by centrifugation at 3,500 g for 1 h at 4 °C. Virus particles were pelleted over a 2 ml, 30% sucrose cushion (in 50 mM HEPES, pH 8.0, containing 200 mM NaCl) at 105,000g for 3 h at 4 °C. Pellets were resuspended in HEPES buffer containing 0.5% (v/v) NP-40 overnight at 4 °C, clarified by centrifugation (16,000g for 10 min at 4 °C), layered onto a 15–45% sucrose gradient and fractionated by centrifugation at 105,000 g for 3 h at 4 °C. Fractions were analysed by SDS–PAGE and subsequent Coomassie staining, and those containing virus particles pooled. Sucrose was removed by desalting with a spin column (Zeba, Pierce) and samples concentrated by ultrafiltration (Amicon).

**Expression and purification of integrin αvβ6.** Soluble αvβ6 headpiece was prepared as described earlier[15]. Briefly, the αv headpiece construct contained residues 1–594 of αv with an M400C mutation followed by a 3C protease site, the ACID coiled coil, a strep II tag and a 6 × histidine tag. β6 headpiece residues 1–474 with an I270C mutation or β3 headpiece residues 1–472 with a Q267C mutation, were followed by a 3C site, BASE coiled coil, and a histidine tag. The mutations generated a disulfide bond creating an α/β heterodimer. Proteins were co-expressed in HEK293S Gnt I − cells[31] with Ex-Cell 293 serum-free medium (Sigma) and purified with Ni-NTA affinity columns (Qiagen). Protein was cleaved by 3C protease at 4 °C overnight, passed through Ni-NTA resin and further purified with an ion-exchange gradient from 50 mM to 1 M NaCl, 20 mM Tris-HCl, pH 8.0 (Q fast-flow Sepharose, GE Healthcare) and finally gel filtrated (Superdex 200, GE Healthcare).

**Crystallographic structure determination of FMDV PanAsia.** Purified inactivated PanAsia particles (virus source Merck Sharp Dohme Animal Health) were concentrated to 2.2 mg ml⁻¹ in HEPES pH 8.0, 200 mM NaCl and crystals grown by the sitting-drop vapour-diffusion method in Crystalquick X plates

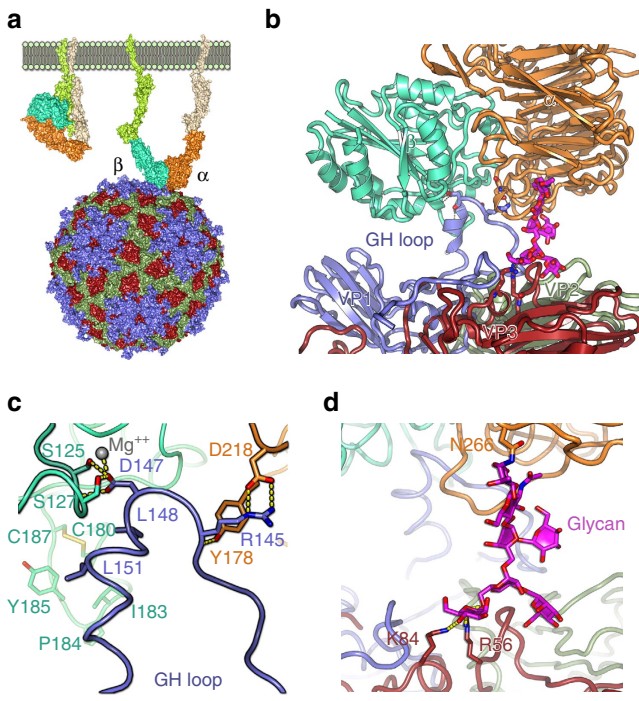

**Figure 5 | The major binding mode.** (**a**) The schematic shows the change in conformation of the integrin from the 'closed' state (left) to the 'open' (right) as visualized in the capsid interaction. (**b**) Side-view showing an FMDV biological protomer drawn in cartoon format with the proteins coloured by chain as in Fig. 1c. The integrin subunits are drawn in ribbon representation and coloured as in Fig. 2. Key interacting residues are drawn as sticks. The sugar is drawn as sticks in magenta. (**c**) Close-up of the VP1 GH loop/integrin interactions shown in **a** with the protein backbone drawn as a ribbon and the interacting side-chains as sticks. The magnesium ion expected to coordinate the MIDAS interaction is included as a grey sphere. (**d**) Close-up of the sugar interactions shown in **a** with the proteins depicted as in **b**.

(Greiner Bio-One) at room temperature (294 K) by dispensing 100 nl virus and 100 nl precipitant with a Cartesian robot as described previously[32]. Initial screening was performed using the SaltRx crystallization screen (Hampton Research). Microcrystals of PanAsia grew within a week with 1.5 M ammonium sulfate, 100 mM bis-Tris propane, pH 7.0. Optimization by varying the concentration of ammonium sulfate around the initial condition produced diffraction quality crystals. Depending on the size of the crystals, either a $50 \times 50$ or $100 \times 100 \, \mu m^2$ beam ($\lambda = 0.9778$ Å; I24 micro-focus beamline, Diamond) was used for *in-situ* diffraction image collection[33] at 294 K on a Pilatus 6 M detector. The crystals belonged to space group I23 with cell dimensions $a = 345$ Å and data were collected to 2.3 Å resolution (Table 1).

The structure was solved by molecular replacement with coordinates and non-crystallographic symmetry (NCS) operators from O1BFS (PDB 1BBT)[34]. Initial phase estimates were obtained by rigid-body refinement with CNS[35]. Iterative positional and B-factor refinement (via CNS) used strict NCS constraints. Phases were further improved by fivefold NCS averaging.

**Complex preparation and cryo-EM data collection.** We formed the FMDV-integrin complex for cryo-EM by incubation of the soluble integrin $\alpha v \beta 6$ headpeice with FMDV PanAsia or O1M overnight at 4 °C, in HEPES buffer containing divalent cations (either 2 mM $Mg^{2+}$ or 2 mM $Mn^{2+}$). The incubations contained five integrin molecules per binding site (assuming 60 binding sites per capsid), with a concentration of integrin of 10 μM and virus 0.033 μM. A 4 μl aliquot of the mixture was applied on a glow-discharged holey carbon-coated copper grid (C-flat, CF-2/1-2C; Protochips), the grid was blotted for 3 s in 70–90% relative humidity and plunge-frozen in liquid ethane using a Vitrobot mark IV (FEI).

Cryo-EM data were collected using a Tecnai F30 'Polara' microscope (FEI) operated at 300 kV, equipped with an energy filter (GIF Quantum; Gatan, Pleasanton, CA) operating in zero-loss mode (20 eV energy selecting slit width), and a direct electron detector (K2 Summit; Gatan). Data were collected as movies (25 frames, each 0.2 s) in single electron counting mode with SerialEM[36]

using a defocus range 1.5–3.0 μm and at calibrated magnification of $\times 37,037$, corresponding to pixel size of 1.35 Å. Dose rate at the detector was 8 e⁻ per pixel per s, resulting in total electron dose of 25 e⁻ Å⁻² in the specimen.

**Image processing and 3D image reconstruction.** Frames from each movie were aligned and averaged to produce drift-corrected micrographs[37] and the contrast transfer function parameters were estimated using CTFFIND3 (ref. 38). Micrographs showing signs of astigmatism or significant drift were discarded. Particles were selected automatically using ETHAN[39] and then visually screened in EMAN2 (ref. 40). Structures were determined with Relion 1.3 following the so-called gold-standard refinement procedure to prevent overfitting[24] with icosahedral symmetry applied. Reference-free two-dimensional (2D) and template-based 3D classification was used to select the most ordered fraction of particles. The X-ray structure of native FMDV O1BFS (PDB:1BBT)[34] was low-pass-filtered to 50 Å resolution and used as an initial template for 3D classification and refinement. A total of 1,649 O1M particles from 235 micrographs and 13,438 PanAsia particles from 359 micrographs were used to calculate the final density maps at 3.5- and 3.1-Å resolution, as estimated by Fourier shell correlation, respectively. Final maps were sharpened by applying an inverse B-factor and the effect of masking[41,42] was taken into account by high-resolution phase randomization. The final refinement parameters were used for extraction of sub-particles for asymmetric reconstruction of the FMDV-integrin complexes.

**Sub-particle extraction and asymmetric reconstruction.** The location and approximate orientation of one integrin molecule was defined as a vector from the centre of the icosahedrally averaged map to the centre of the integrin density (at radius of 144 Å) in UCSF Chimera[43]. The locations of all 60 symmetry-related sub-particles (potential integrin positions) were then calculated using the localized reconstruction method[23] and the sub-particles were extracted. Before extracting the sub-particles from the images, the density corresponding to the genome and the capsid were subtracted (based on the icosahedrally averaged reconstruction). A spherical mask (diameter 128 Å) defining roughly one integrin density in the icosahedrally averaged map was defined and icosahedral symmetry was applied to this mask. The sub-particles were extracted into individual boxes and densities outside the spherical mask were subtracted using *relion_project*. Extracted sub-particles were then subjected to 3D classification in Relion. To balance the number of sub-particles per class and separation of sub-particles into distinct classes, 3D classification was first tested using different numbers of classes. Classification in 20 classes was used as it provided the clearest class averages for the distinct poses of the integrin. An initial model was created from the extracted sub-particles using *relion_reconstruct*[44] and used as a template for 3D classification. To avoid biasing the classification due to the relatively weak signal in the sub-particles, no changes in the orientations or origins were allowed and only information to 12 Å resolution was included for the classification. To limit the effect of high frequencies further, only phase flipping was performed during CTF-correction. The final resolution of the best classes was assessed by Fourier shell correlation (threshold 0.143) calculated between reconstructions generated from two independent half-sets of the sub-particles[41].

**Analysis of integrin distributions on the particles.** For each of the analysed binding modes, the contributing integrins were mapped back onto the virus particles to analyse whether binding increases or decreases the probability of neighbouring sites being occupied. Based on the relative locations of the binding sites, the theoretical probability of any neighbouring site being occupied can be calculated. For example for the 2-fold related integrins, the probability of finding exactly k such pairs on a capsid with n bound integrins is given by the following formula involving binomial coefficients:

$$2^{(n-2k)} \times \binom{30 - k}{n - 2k} \times \binom{30}{k} \div \binom{60}{n}$$

This analysis was repeated for each tuple $(n, k)$, where $0 < n \leq 60$, $0 < 2k \leq n$. The calculated theoretical probabilities were compared to the observed occupancies.

**Model building and refinement.** The crystallographic structure of O1M (PDB:5DDJ (ref. 30)) was fitted in the EM density as a rigid body with UCSF Chimera[43]. The fitting was further improved by positional and B-factor refinement in real space with Phenix[45] and COOT[46], iteratively. For O Panasia, the crystal structure obtained here, built using COOT[46] was used as a template and iteratively refined as above. Each round of model optimization was guided by cross-correlation between the map (which was kept constant) and the model. Refinement statistics are given in Table 2. For fitting of the integrins, only rigid-body refinement of the entire molecule was used to avoid overfitting[45–47].

**Data availability.** The atomic coordinates and structure factors for the X-ray structure of FMDV Panasia have been submitted to the Protein Data Bank with accession number PDB: 5NE4. The cryo-EM density maps of the FMDV Panasia

and O1M complexes with αvβ6 have been deposited with the Electron Microscopy Data Bank (associated PDB code are given in parentheses): Panasia virus, EMD-3630 (PDB:5NED); Panasia plus integrin Pose A, EMD-3632 (PDB:5NEM), Panasia plus integrin Pose A', EMD-3633 (PDB:5NER); O1M virus, EMD-3631 (PDB:5NEJ); O1M plus integrin Pose A, EMD-3634 (PDB:5NET), O1M plus integrin Pose B, EMD-3635 (PDB:5NEU). All other relevant data are available from the authors upon request.

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

## Acknowledgements

We are thankful to Robert Esnouf and Jun Dong for IT support; beamline scientists at the Diamond Light Source for assistance. The OPIC electron microscopy facility was founded by a Wellcome Trust JIF award (060208/Z/00/Z) and is supported by a WT equipment grant (093305/Z/10/Z). This is a contribution of the Oxford Instruct Centre. N.G.A.A. is supported by the Spanish Ministerio de Economia y Competitividad (BFU2015-64541-R), J.T.H. by the European Research Council under the European Union's Horizon 2020 research and innovation programme (649053), A.K. by Wellcome Trust, Q.W. and R.Z. Chinese Academy of Sciences and D.I.S. and E.E.F. by the UK MRC (G1000099, G1100525/1 and MR/N00065X/1) and the Wellcome Trust (090532/Z/09/Z).

## Author contributions

A.K., Q.W., X.D., M.O. and J.S. prepared samples, A.K., Q.W., S.L.I., J.T.H., E.E.F., N.G.A.A. and D.I.S. analysed the data. A.K., B.C., R.Z., E.E.F., T.A.S., N.G.A.A., J.T.H. and D.I.S. designed the study and with S.L.I. wrote the manuscript.

## Additional information

**Competing interests:** The authors declare no competing financial interests.

**Publisher's note**: 

