## [Peer Review File · Nature Communications]

Reviewers' comments:

Reviewer #1 (Remarks to the Author):

FMDV is an important pathogen worldwide. It is accepted that the antigenic GH loop of the viral capsid protein VP1 mediates cell entry by attachment to an integrin receptor and acidic conditions in the endosome are thought to trigger particle uncoating. Some variants have adapted to use HS for entry and the HS binding site has been mapped. The authors have solved the high resolution cryo-EM structures of $\alpha\beta 6$ and two tissue culture adapted FMDV strains. They used a powerful localised reconstruction method to sort and classify bound integrin in different poses and visualize the conformational changes to the VP1 GH loop.

The sections describing extended data figures 6-7 are particularly dense and could use expansion or clarification. As it stands, a reader must access the extended data and digest the figure legends to follow the text. Nevertheless, an excellent work with beautiful structures and nice use of localized reconstruction to tease out the different conformations of the bound integrin.

Minor:

Extended Data Figs 6–8 are called when a fit of a structure into density (not the footprints in Fig 8) is discussed.

Please indicate the location of the additional (unfilled) density observed in pose A to bridge the integrin to the virus surface in Extended Data Fig 6. Similarly, where is the N-linked sugar at N266 of the $\alpha\beta$ -propeller domain? Some arrows would be helpful to the reader.

Extended Data Fig 5 (a) is not mentioned in the text. Extended Data Fig 5 and 6 seem too important to be relegated to Extended Data.

Extended Data Fig 8 is not discussed or called (except as noted above regarding the fit of the helix into the loop density). Fig. 3 d is not called.

A symbol did not translate into correct font in the section in Methods on Expression and purification of integrin $\alpha\beta 6$

The only mention of a high resolution X-ray structure is one sentence in the text to describe the GH loop as the only conformational change.

In Methods, "3D classification into twenty classes in Relion (twenty was chosen by trail and error)" begs the question: what was better about 20 compared to 15 or 25?

"For O Panasia (PanAsia), O1M was used as a template and the correct sequence built using COOT41 followed by iterative refinement" why was this approach used instead of using the crystal structure of PanAsia solved and presented in the manuscript?

Reviewer #2 (Remarks to the Author):

In this paper the authors describe cryoEM studies that attempt to define the interaction of foot and mouth disease virus (FMDV) with its cellular integrin receptor. The analysis is not straightforward as the binding sites on the virus are not saturated and the integrin adopts multiple settings. This means that simple icosahedral reconstruction of the complex leads to a very blurred density for the integrin moiety. The authors therefore use a method that some of them developed earlier to mask out and classify the localized region (sub-particle) around each potential binding site on the virus in the expectation that this more focussed analysis will yield a clearer map of the binding site and its integrin binding partner. Using this approach the authors analyse two serotypes of FMDV in complex with the head part of integrin $\alpha\beta 6$. Even with the localized reconstruction, the density for the integrin is poorly resolved but the overall shape is sufficiently well defined to fit in the atomic model of integrin from a crystal structure. This allows the authors to propose binding models for the integrin on the two FMDV serotypes.

The work is technically competent and mostly well described. The results will be of interest to virologists, particularly those concerned with FMDV. A number of points require the authors' attention.

p.4/5 It must be of concern that even with the sub-classification only 36% of occupied sites in PanAsia and 20% of occupied sites in O1M are assigned to the classes that the authors interpret. Reasons could be that the assigned classes correspond to "end points" of the movement of the RGD loop, with everything in between smeared out. Also the "sub-particles" in this system are quite crowded and there will be substantial overlap between them in projection when the sub-particle data are extracted from the original images. This means that the densities ascribed to any sub-particle from the different views will not be consistent, thus interfering with the reconstruction. The authors should address these problems.

p.2 l.-2 The RGD motif is later referred to in text and figures as residues 145-147, not 141-143. There appears to be inconsistent numbering.

p.3 l.4 Clearer to say: "An α and β domain come together...", otherwise it sounds like a tetramer.

p.3 l.14 Helpful to add: ...residues L148 and L151 at D+1 and D+4 are key...

p.4 l.10 Helpful to add: ...a high resolution X-ray structure determined here of native PanAsia.... Otherwise the fact that a crystal structure has been solved gets rather lost.

p.4 l.15 The deconvolution procedure does not 'account for' the incomplete occupancy, which must arise from inadequacy in the preparation. Better to say: ...modes of integrin attachment, despite the variable occupancy of the integrin.

p5. l.10 ...bound to pro-TGF-b1. Give reference.

p.9 l.1 Proteins were co-expressed...

p.10 l.-3 Reference 33 (Chen et al.) is not appropriate for drift correction but is appropriate for phase randomization (p.11).

p.12 l.1 What criterion was used to select 20 classes, as opposed to some other number, on the basis of trial and error?

Fig.1d The N-terminus of the RGD loop does not appear to start from the same place on the capsid in the blue and cyan versions, although the C-termini do appear to coincide. Presumably if it is described as 'a loop' the starts and ends should agree.

p.20 l.12 ...two leucines (L148 and L151), not L184

Extended data fig.5. In panels c and d the enclosed map volume for the distal part of the green integrin chain appears much larger than the modelled domain. Do the authors have an explanation for this?

Extended data fig.8. In my printout the residue labels are illegible. Would it be better to show a smaller area at higher magnification?

Rules of engagement between $\alpha\beta6$ integrin and the RGD-loop of foot-and-mouth disease virus
Kotecha et al

Reviewer #1 (Remarks to the Author):

FMDV is an important pathogen worldwide. It is accepted that the antigenic GH loop of the viral capsid protein VP1 mediates cell entry by attachment to an integrin receptor and acidic conditions in the endosome are thought to trigger particle uncoating. Some variants have adapted to use HS for entry and the HS binding site has been mapped. The authors have solved the high resolution cryo-EM structures of $\alpha\beta6$ and two tissue culture adapted FMDV strains. They used a powerful localised reconstruction method to sort and classify bound integrin in different poses and visualize the conformational changes to the VP1 GH loop.

The sections describing extended data figures 6-7 are particularly dense and could use expansion or clarification. As it stands, a reader must access the extended data and digest the figure legends to follow the text. Nevertheless, an excellent work with beautiful structures and nice use of localized reconstruction to tease out the different conformations of the bound integrin.

We would like to thank the reviewer for their encouraging comments. We have now brought Extended Data Figs 5 and 6 into the main figures, to make the paper easier to follow.

Minor:

Extended Data Figs 6–8 are called when a fit of a structure into density (not the footprints in Fig 8) is discussed.

This call-out has been corrected to exclude what was Extended Data Fig 8.

Please indicate the location of the additional (unfilled) density observed in pose A to bridge the integrin to the virus surface in Extended Data Fig 6. Similarly, where is the N-linked sugar at N266 of the $\alpha\beta$ -propeller domain? Some arrows would be helpful to the reader.

This figure has been annotated to show the position of N266 by an orange star and the attached sugar (which fills the additional density) is shown in magenta. In this view the separation between the densities is not clear. The legend has been correspondingly updated.

Extended Data Fig 5 (a) is not mentioned in the text. Extended Data Fig 5 and 6 seem too important to be relegated to Extended Data.

We agree and have made these main figures 3 and 4 with specific mention of Fig 3 (a) in the text.

Extended Data Fig 8 is not discussed or called (except as noted above regarding the fit

of the helix into the loop density). Fig. 3 d is not called.

An appropriate call out has been included for Extended Data Fig 6/7 (as renumbered) and Fig 5d (as renumbered)

A symbol did not translate into correct font in the section in Methods on Expression and purification of integrin $\alpha\beta6$

We will try to make sure this gets carried through to the pdf correctly.

The only mention of a high resolution X-ray structure is one sentence in the text to describe the GH loop as the only conformational change.

This has been amended in the introductory section and on P.4.

In Methods, “3D classification into twenty classes in Relion (twenty was chosen by trail and error)” begs the question: what was better about 20 compared to 15 or 25?

We have modified the sentence as follows: “Extracted sub-particles were then subjected to 3D classification in Relion. To balance the number of sub-particles per class and separation of sub-particles into distinct classes, 3D classification was first tested using different numbers of classes. Classification in 20 classes was used as it provided the clearest class averages for the distinct poses of integrin part.”

“For O Panasia (PanAsia), O1M was used as a template and the correct sequence built using COOT41 followed by iterative refinement” why was this approach used instead of using the crystal structure of PanAsia solved and presented in the manuscript?

At the time that the template was required the model for PanAsia had not been refined. This has subsequently been repeated with the correct model. The text has thus been updated.

Reviewer #2 (Remarks to the Author):

In this paper the authors describe cryoEM studies that attempt to define the interaction of foot and mouth disease virus (FMDV) with its cellular integrin receptor. The analysis is not straightforward as the binding sites on the virus are not saturated and the integrin adopts multiple settings. This means that simple icosahedral reconstruction of the complex leads to a very blurred density for the integrin moiety. The authors therefore use a method that some of them developed earlier to mask out and classify the localized region (sub-particle) around each potential binding site on the virus in the expectation that this more focussed analysis will yield a clearer map of the binding site and its integrin binding partner. Using this approach the authors analyse two serotypes of FMDV in complex with the head part of integrin $\alpha\beta6$. Even with the localized reconstruction, the density for the integrin is poorly resolved but the overall shape is sufficiently well defined to fit in the atomic model of integrin from a crystal structure. This allows the authors to

propose binding models for the integrin on the two FMDV serotypes.

The work is technically competent and mostly well described. The results will be of interest to virologists, particularly those concerned with FMDV. A number of points require the authors' attention.

p.4/5 It must be of concern that even with the sub-classification only 36% of occupied sites in PanAsia and 20% of occupied sites in O1M are assigned to the classes that the authors interpret. Reasons could be that the assigned classes correspond to “end points” of the movement of the RGD loop, with everything in between smeared out. Also the “sub-particles” in this system are quite crowded and there will be substantial overlap between them in projection when the sub-particle data are extracted from the original images. This means that the densities ascribed to any sub-particle from the different views will not be consistent, thus interfering with the reconstruction. The authors should address these problems.

The reviewer raises a very interesting point that the poses of integrin we observed might be end-points of a continuum of different poses. If this is the case, then either the 3D classification and localized reconstruction methods we used were not sensitive enough or the number of particles was too small, to resolve these intermediate conformations.

We agree with the reviewer that the projected integrin densities also overlap each other. As clear class assignments and thus orientations are known only for a fraction of the integrins, overlapping integrin densities cannot be subtracted from the projection image. The overlapping densities are likely to obscure the 3D classification, in addition to a possible continuum of different poses.

We have added the following to explicitly mention these possible caveats: “Only a small fraction of integrin sub-particles was assigned to a particular pose. It is possible that some of the unassigned sub-particles correspond to different poses of the integrin that were not detected in our classification. The sub-particles also unavoidably overlap each other in the projection image, further hampering their 3D classification (Ilca et al. 2015). Likely due to these limitations, integrin density was resolved to a lower resolution than the capsid.”.

p.2 l.-2 The RGD motif is later referred to in text and figures as residues 145-147, not 141-143. There appears to be inconsistent numbering.

Thank you – this has been corrected.

p.3 l.4 Clearer to say: “An α and β domain come together...”, otherwise it sounds like a tetramer.

Thanks you, the text has been changed as suggested.

p.3 l.14 Helpful to add: ...residues L148 and L151 at D+1 and D+4 are key...

Added.

p.4 l.10 Helpful to add: ... a high resolution X-ray structure determined here of native PanAsia.... Otherwise the fact that a crystal structure has been solved gets rather lost.

Yes....this has been added.

p.4 l.15 The deconvolution procedure does not 'account for' the incomplete occupancy, which must arise from inadequacy in the preparation. Better to say: ...modes of integrin attachment, despite the variable occupancy of the integrin.

This has been re-phrased as suggested.

p5. l.10 ...bound to pro-TGF-b1. Give reference.

This has been inserted.

p.9 l.1 Proteins were co-expressed...

Corrected

p.10 l.-3 Reference 33 (Chen et al.) is not appropriate for drift correction but is appropriate for phase randomization (p.11).

The correct reference is now given Li X et al., 2013.

p.12 l.1 What criterion was used to select 20 classes, as opposed to some other number, on the basis of trial and error?

Please see our answer to Reviewer 1.

Fig.1d The N-terminus of the RGD loop does not appear to start from the same place on the capsid in the blue and cyan versions, although the C-termini do appear to coincide. Presumably if it is described as 'a loop' the starts and ends should agree.

This has been corrected so that the loops originate at the same point. This panel has been redrawn and we believe it is now clearer.

p.20 l.12 ...two leucines (L148 and L151), not L184

Corrected.

Extended data fig.5. In panels c and d the enclosed map volume for the distal part of the green integrin chain appears much larger than the modelled domain. Do the authors have an explanation for this?

This density comes from symmetry related integrin molecules.

Extended data fig.8. In my printout the residue labels are illegible. Would it be better to show a smaller area at higher magnification?

We have included an extra Extended Data figure to supplement this (Extended data Figure 7 as renumbered) to show a smaller area at higher magnification, the lower magnification pictures are retained to establish the context.

REVIEWERS' COMMENTS:

Reviewer #1 (Remarks to the Author):

All previous concerns have been adequately addressed.

Rules of engagement between $\alpha v\beta 6$ integrin and the RGD-loop of foot-and-mouth disease virus

Response to referees' comments

We thank the reviewers for their second review. There are no points to address:

Reviewer #1 (Remarks to the Author):

All previous concerns have been adequately addressed.